# Improving Glioma Segmentation in Low-Resolution Domains with Transfer Learning

**Juampablo Enrique Heras Rivera**[1]                                             JEHR@UW.EDU
[1] *Mechanical Engineering, University of Washington, Seattle, WA, USA*

**Harshitha Rebala**[2]                                                         LHREBALA@UW.EDU
[2] *Paul G. Allen School of Computer Science, University of Washington, Seattle, WA, USA*

**Tianyi Ren**[1]                                                                 TR1@UW.EDU
**Abhishek Sharma**[1]                                                           AS711@UW.EDU
**Mehmet Kurt**[1]                                                               MKURT@UW.EDU

**Editors:** Under Review for MIDL 2024

## Abstract

Training accurate tumor segmentation models only using data from the BraTS Sub-Saharan Africa (SSA) Glioma dataset is difficult due to the low quantity and resolution of the images. However, it is possible to improve model performance through the use of transfer learning methods which leverage insights gained from larger datasets, such as the BraTS23 Adult Glioma dataset. Here, we evaluate the performance of various transfer learning approaches on the task of improving tumor segmentation Dice and Hausdorff Distance (95%) scores on the BraTS SSA dataset. The transfer learning approaches assessed here include: Domain Adversarial Neural Networks, Fine Tuning (with and without freezing layer weights), and training with a combined dataset of low- and high-resolution images.

**Keywords:** Tumor segmentation, transfer learning, domain adaptation, low-resolution tumor segmentation.

## 1. Introduction

The performance of deep learning models depends on the quality of training data used, so models trained on insufficient or low-quality data will tend to perform poorly. This tendency is exemplified in the variation in performance between models trained using the BraTS23 Sub-Saharan Africa (SSA) dataset and the BraTS23 Adult Glioma dataset. When evaluated on their respective datasets, models trained with the BraTS23 Adult Glioma dataset significantly outperform their SSA counterparts. To understand this disparity, it is important to consider the differences between the two datasets. Both datasets contain brain MRI scans of adult glioma patients with radiologist annotations of tumorous subregions; however, the BraTS23 Adult Glioma dataset consists of 1126 brain MRI scans, while the BraTS SSA dataset consists of 95 scans. Notably, the datasets also differ in resolution, as the MRI scanners used in the Brats23 Adult Glioma dataset are of 3T magnetic field strength, while the scanners in the BraTS SSA dataset are of 1.5T magnetic field strength. Furthermore, patients in the BraTS SSA dataset are typically scanned later in the disease

progression at more advanced stages, and present the unique characteristics of gliomas in SSA (e.g., suspected higher rates of gliomatosis cerebri)(Adewole et al., 2023).

Machine learning methods in the field of transfer learning provide a way to bridge the gap in segmentation quality between models trained on the BraTS23 Adult Glioma and SSA datasets by "transferring" insights gained from a larger domain (or dataset in this instance) to a smaller one. To this end, we assess multiple transfer learning frameworks including Domain Adversarial Neural Networks (DANN) (Ganin et al., 2016), fine tuning models pre-trained on the BraTS Adult Glioma dataset to the BraTS SSA dataset, and training with an augmented dataset consisting of samples from both domains.

## 2. Methods

### 2.1. Experiments

We evaluated the validation performance on the BraTS SSA dataset of the following 6 approaches, all based on the U-Net architecture described in (Ren et al., 2024):

| | |
|---|---|
| **Baseline (BraTS Adult Glioma)** | Train the U-Net model on the complete BraTS23 Adult Glioma dataset. This serves as a baseline to evaluate the efficacy of the transfer learning approaches. |
| **Combined Dataset Training** | Train the U-Net model on a combined dataset consisting of both the BraTS SSA data and BraTS23 Adult Glioma data. |
| **Fine-Tuning** | Start with the trained baseline model, then fine-tune the entire model (all layers) by continuing training on the BraTS SSA data. |
| **Fine-Tuning (Frozen Decoder)** | Similar to Fine-Tuning, but freeze the decoder layers during the second stage of training. |
| **DANN** | Train a combined BraTS23 Adult Glioma and SSA dataset with DANN architecture to create a model that learns domain-invariant features useful for segmentation across both datasets. |
| **DANN without gradient reversal** | To explore the effect of gradient reversal, train an identical model to the DANN, but without the gradient reversal layer. |

Table 1: Descriptions of transfer learning approaches considered.

### 2.2. Implementation and Training Schedule

The experiments were conducted in PyTorch using 2 NVIDIA A40 GPUs with 40 GB of memory to train each model. During training, the Adam optimizer was used with a decaying learning rate starting at $\alpha_0 = 6 \times 10^{-5}$, and subsequent learning rates calculated as follows: $\alpha_i = \alpha_0 \times \left(1 - \frac{\text{epoch}_i}{\text{epoch}_N}\right)^{0.75}$, where $\alpha_i$ is the learning rate at epoch $i \in 1, \ldots, N$. The model was trained using a batch size of 1 due to the large size of the training sam-

ples. To evaluate the models' generalizability, we performed 10-fold cross-validation for all experiments (approximately an 85-10 subject training/validation split).

An optimized U-Net (Ren et al., 2024) was used as the backbone model in all approaches. The model inputs for each subject are four different MRI contrasts, namely native (T1), post-contrast T1-weighted (T1Gd), T2-weighted (T2), and T2 Fluid Attenuated Inversion Recovery (T2-FLAIR). The outputs of the model are 3 channel images containing segmentation maps of each tumor sub-region (NCR, ED, ET). For the DANN approach, binary cross entropy (BCE) was used at the classifier branch, and a weighted combination of Mean Squared Error (MSE) and Cross Entropy (CE) was used for segmentation loss. The total backpropagated loss is: $\mathcal{L} = \mathcal{L}_{seg} + \mathcal{L}_{class}$, where $\mathcal{L}_{seg}$ and $\mathcal{L}_{class}$ are the segmentation and classification losses, respectively, with a 1/50 scaling on the classification loss to ensure the two losses are of similar magnitude.

## 3. Experiments

| Method | Mean Dice | | | | Mean HD | | | |
|---|---|---|---|---|---|---|---|---|
| | NCR | ED | ET | Average | NCR | ED | ET | Average |
| Baseline (BraTS23 Adult Glioma) | 0.519 | 0.726 | 0.701 | 0.649 | 17.019 | 15.791 | 13.672 | 15.494 |
| Fine-Tuning | 0.829 | 0.896 | 0.879 | 0.867 | 3.851 | 3.454 | 4.257 | 3.854 |
| Fine-Tuning (Frozen Decoder) | 0.792 | 0.881 | 0.860 | 0.845 | 4.483 | 3.931 | 6.886 | 5.100 |
| Combined Dataset Training | 0.836 | 0.899 | 0.879 | 0.871 | 3.135 | 2.606 | 3.393 | **3.045** |
| DANN | **0.847** | 0.897 | **0.893** | **0.879** | **3.076** | 3.828 | **3.219** | 3.374 |
| DANN without gradient reversal | 0.842 | **0.902** | 0.892 | 0.878 | 4.951 | **2.345** | 4.421 | 3.906 |

Table 2: Mean Dice and HD-95 values of 10-fold cross-validation for the transfer learning approaches considered. The best scores for each category are bolded. NCR: necrotic tumor core, ED: peritumoral edema, ET: enhacing tumor.

Table 2 shows the Dice coefficient and HD95 score results of three prediction labels and their average from 10-fold cross-validation analysis. From Table 2 we see that all of the transfer learning methods significantly outperform the baseline method with a 20% of improvement in Dice. Additionally, it is observed that the DANN model outperforms other domain adaptation methods in NCR and ET for both Dice and HD95 scores. To establish statistical significance of these results, a paired sample t-test was performed on the cross-validation results. The t-test revealed significance in the differences between the DANN-based training approaches and the Combined Dataset training approach. Specifically, DANN demonstrates superior performance compared to the Combined Dataset Training in terms of average Dice ($p=0.0506$) and average HD-95 ($p=0.0236$), while the DANN without gradient reversal approach shows improvement in Dice ($p=0.0519$).

## 4. Conclusion

Following an analysis of domain adaptation methods aimed at improving the quality of medical image segmentation in low-resolution datasets, we have preliminary support encouraging the use of the DANN framework in future projects where domain adaptation is required.

## Acknowledgments

The work of Juampablo Heras Rivera was partially supported by the U.S. Department of Energy, Office of Science, Office of Advanced Scientific Computing Research, Department of Energy Computational Science Graduate Fellowship under Award Number DE-SC0024386.

The authors are especially grateful to Dr. Udunna Anazodo and Maruf Adewole for their contributions to this project.

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
