# OpenReview forum: "Improving Glioma Segmentation in Low-Resolution Domains with Transfer Learning"
_MIDL.io/2024/Short_Papers — MIDL 2024 Short Papers_

### Official Review · Reviewer_T7xY · 2024-04-24

**Confidence:** 3
**Final Rating:** 3.5

**Review:**

This paper evaluates various transfer learning approaches for brain tumor segmentation using two glioma datasets (one large and one small dataset). Techniques considered are joint training, fine-tuning (with and without freezing layers) and domain adversarial approaches. Domain adaptation techniques have been throughly studied in various medical image analysis context; therefore, the contribution of this study is limited; however, it could be of interest to researchers specifically interested in brain tumor segmentation. The paper is not clear about what is meant by domain adversarial neural networks (DANN) without gradient reversal. This might not be immediately clear to all readers including this reviewer. If it simply means removing the gradient reversal layer from DANN, the highly accurate results obtained with this method are surprising and should merit more discussion as gradient reversal is a key element in DANN.

---

### Decision · Program_Chairs · 2024-04-26

Accept